# Flora of Algae and Cyanobacteria of Continental Waters of Israel in the XXI Century: Taxonomy, Autecology and Water Quality Indicators

Sophia Barinova [1,*] and Thomas Smith [2]

1  Institute of Evolution, University of Haifa, Abba Khoushi Ave, 199, Mount Carmel, Haifa 3498838, Israel
2  Division of Math and Science 1000 W Iowa St, Arkansas State University Beebe, Beebe, AR 72012, USA; tesmith@asub.edu
*  Correspondence: sophia@evo.haifa.ac.il; Tel.: +972-4824-97-99

**Abstract:** The article presents a list of algal species and cyanobacteria found in the continental waters of Israel in 1898–2022. Research progressed in 2000–2022 by increasing the list from 1261 to 1628 species belonging to fourteen phyla. Taxonomic analysis shows that diatoms, cyanobacteria, and green algae predominated. The first-time data has been synthesized to identify the indicator properties of Israel's aquatic flora carried out on algae and cyanobacteria, which can be used to monitor water quality. The species' ecological preferences are given for ten environmental variables: substrate preference, temperature, oxygen saturation with water mobility, water pH preferences, water salinity, organic pollution according to Watanabe and Sládeček with species-specific index of saprobity S, trophic state, and type of nutrition (autotrophic or heterotrophic). This list of species with indicator values for each species is used to characterize the water properties in Israel. In addition, it can be applied to assess the state of aquatic ecosystems and monitor water quality based on bioindication methods.

**Keywords:** algae; diatom; green; cyanobacteria; flora; ecology; bioindicators; Israel





## 1. Introduction

Studying the diversity of algae is not an easy task. The list of species of the so-called alpha diversity can be compiled as a result of many years of research in a particular region and habitat characterization. At the same time, identified species of algae can be indicators of water quality if we know the ecological preferences of each species [1,2]. Thus, the identification of the diversity of organisms in aquatic habitats is not only the task of floristic research but also the basis for subsequent ecological conclusions, because algae and cyanobacteria react quickly to changes in the environment in the aquatic ecosystem, and also, due to their high diversity, they are used worldwide as bioindicators for assessing water quality [1–3]. In some cases, floristic works compile only a list of species [4,5], but most often the authors provide limited information about the ecological preferences of specific species [6–8], which can be used as indicators of some parameters of the environment or give descriptive results for some parameters of the studied flora [1,8]. The most developed ideas are about the indicator properties of diatoms [9,10]. However, diatoms represent only about half of the species composition in aquatic communities. There are still very few such publications, which simultaneously include a list of both diatom and non-diatom species, as well as all possible indicator properties of algae and cyanobacteria [6,7].

One of the main tasks of the monitoring system is not only to assess water quality but also to identify sources of pollution. Most water comes from natural sources, including rivers, lakes, and reservoirs. Consequently, the quality of water in them should be assessed and predicted, taking into account changes in anthropogenic pressure and climate. Water quality is formed in natural conditions and depends on the river basin and the reservoir

ecosystem. Considering that water quality assessment requires high costs, the development of rapid methods is an urgent task.

Determination of water quality using bioindicators is based on the conformity of the ecology of species in the community to the environment in which they exist. The relationships between different levels of organisms are based on the hierarchical organization of the biotic community and are described by the trophic pyramid model [11,12]. The distribution of organisms or species groups over the intervals of environmental factors is of paramount importance in bioindication. Assessing contamination of freshwater sources is challenging, and appropriate methods must be selected carefully. Methods and indicators that can be used to evaluate the impact of pollution on natural water bodies can be both chemical and biological [13,14]. The latter, in turn, are divided into bioindication and biotesting [15]. Suppose the biotest method evaluates the influence of the environment on the test organism in which it is placed. Consequently, the indicator method is based on the ecological character of the relationship between water and biota in an ecosystem [16].

The adaptation level of species determines the species composition in a given aquatic environment [17]. Therefore, with the help of the species composition that exists in a given environment, the environmental variables intervals can be defined by the bioindication method. However, determining the role of specific environmental variables and predicting the community's response to changes in the environment is, in any case, a complex problem. Therefore, it is necessary to collect information on the chemical parameters of water, the species composition, and the abundance of organisms inhabiting it. These two sets of data should then be classified and used in assessing water quality.

Two different aspects of using the information on algae species and their ecology in this article may be helpful to researchers and in monitoring practice. The first aim was data that were collected on the diversity of algae and cyanobacteria inhabiting the reservoirs of Israel, which represents a list of aquatic flora in the current state. Secondly, the established ecological properties of the identified species can be used as bioindicators of water quality in the monitoring system.

This work aimed to compile a list of species from available publications and our recent articles and books on algae and cyanobacteria found in the continental waters of Israel. Based on this list of species and their ecological preferences, we aim to provide not only modern floristic checklist, but also a list of water quality indicators for ten ecological variables.

**Highlights**

- The algae and cyanobacteria flora of continental in Israel are represented.
- First study representing bioindicator data for algae and cyanobacteria in Israel.
- All revealed species of algae and cyanobacteria are indicators of freshwater quality.
- Species list with ecological preferences can be used for monitoring water quality.

## 2. Material and Methods

The first step was to compile the species list from previous publications in 1898–2000 and recent research results in Israel from 2001–2022. The second step was unifying all taxa names under the modern system, especially for the lists of species published in previous periods, to exclude synonyms and double species names. Finally, the third step was creating a system of algae indicators of the continental waters of Israel by combining the list of identified species with the known ecological properties of the species.

Data on the species of algae and cyanobacteria found in the continental waters of Israel were collected from articles published from 1898 to 2000 and summarized with some additions in the book [18].

The recent publications come from algae floristic research [19–77].

The lists of species were also excavated from hydrobiological studies of continental waters [37,41,56,61,62,73,74,78–92].

We summarized information about algal and cyanobacterial species' ecological preferences from more than one hundred published articles and monographs and presented it in

a series of published books [7,20,34] according to major indicator systems for assessment of water pH [93], salinity [94], organic pollution [10,95], nutrition type, and trophic state [9] recommended by EU for biological monitoring [96,97].

The old names of taxa in the lists of Israeli algae species, which were published in different years according to the taxonomy of the publication period, were unified according to the modern system using algaebase.org [97] to compile a complete list of species, excluding synonyms. Then, according to the combined data, the species richness of each taxonomic phylum was analyzed. The ecological properties of identified algae and cyanobacteria in Israel were added to the list of species. The ranges of ecological preferences of each species were divided into categories of bioindicators for the following water variables: pH, salinity, temperature, mobility, and oxygenation of water masses, trophic state, preference for the type of feeding (autotrophic or heterotrophic), and saprobity in two different systems. The distribution of indicator taxa number in each ecological group was ordered to increase the indicated variable value. The standard deviation (STDEV) was calculated for each distribution to identify the richest taxonomic and ecological groups since the STDEV line cuts off more than 50% of the species of the distribution. This can identify successful ecological groups and indicator species in Israel's climatic and anthropogenic conditions. Checking the identified species composition of Israel algae for completeness of the study was carried out by plotting the Willis curve [98], where the distribution of the number of species by the number of genera can be conformed to a logarithmic trend line in the case of completeness of the species list.

## 3. Results and Discussion

As a result, the list of Israel's algae and cyanobacteria was enriched by about four hundred species during the last twenty years. Therefore, floristic species content raised from 1261 species (in 2000, before modern taxonomy update) [11] to 1628 species (in 2022 with modern taxonomy) (Table 1; Table S1). Table 1 does not include eight species with the unclear taxonomic position in the present time, and some mentioned in the references generic names without species definition. As can be seen, diatom species prevail with cyanobacteria and green algae, which cut off by standard deviation line (Figure 1a).

The species list completeness analysis was carried out for this large number of species identified in such small area as Israel, where there are a small number of lakes and small streams due to the peculiarities of the climate. To achieve this, we calculated the distribution of the number of species by the number of genera [98] according to Willis's law (Figure 1b). Since the trend line practically coincides with the distribution line, this suggests that the revealed diversity of algae and cyanobacteria in the aquatic environment of Israel is close to saturation.

**Table 1.** Taxonomic content of algae and cyanobacteria flora in continental aquatic habitats of Israel.

| Phylum | No of Species |
|---|---|
| Bacillariophyta | 535 |
| Cyanobacteria | 432 |
| Chlorophyta | 342 |
| Euglenozoa | 115 |
| Charophyta | 112 |
| Miozoa-Dinophyceae | 27 |
| Ochrophyta-Xanthophyceae | 18 |
| Ochrophyta-Chrysophyceae | 15 |
| Cryptophyta | 10 |
| Ochrophyta-Eustigmatophyceae | 9 |
| Rhodophyta | 7 |
| Haptophyta | 2 |
| Choanozoa | 1 |
| Eukaryota unassigned phylum | 3 |
| Total: | 1628 |

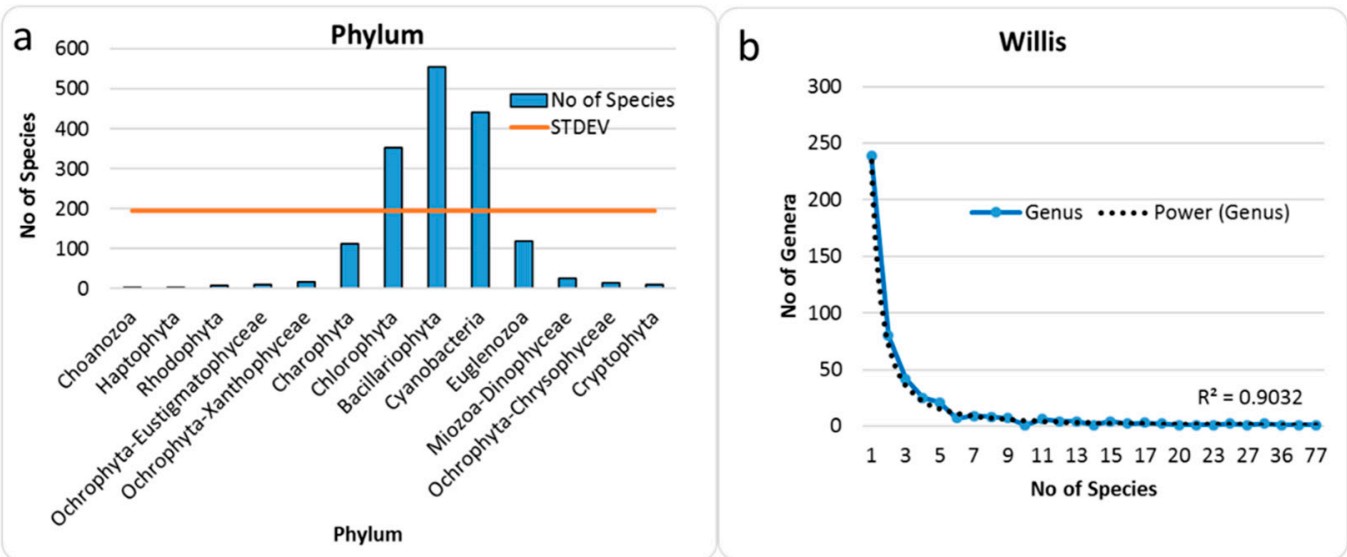

**Figure 1.** Distribution of species richness of algae and cyanobacteria of Israel over taxonomic phyla with STDEV as standard deviation line (**a**), and in the Willis curve as number of genera over species number (**b**).

Species distribution in Figure 1 and Supplementary Table S1 show that in continental flora of algae and cyanobacteria of Israel, three phyla significantly prevail: Bacillariophyta, with 535 species, Cyanobacteria, with 431, and Chlorophyta, with 341 species, of a total 1628 species and 14 phyla (Table 1).

The most species-rich genera of algae and cyanobacteria (Table 2) contain 338 species (about 20% of the total species list) and characterize a flora face. Most of the richest genera belonged to diatoms (162), but next are euglenoids with 75 species (Table 2), contrary to total species richness distribution (Table 1; Table S1). The species-rich genera in Cyanobacteria, Chlorophyta, and Charophyta are significantly less represented in the flora of Israel.

**Table 2.** Most species rich genera of algae and cyanobacteria flora in aquatic habitats of Israel.

| Phylum | Genus | Species |
|---|---|---|
| Bacillariophyta | *Nitzschia* | 68 |
| Bacillariophyta | *Navicula* | 49 |
| Bacillariophyta | *Gomphonema* | 27 |
| Bacillariophyta | *Tryblionella* | 18 |
| | Total | 162 |
| Euglenozoa | *Euglena* | 29 |
| Euglenozoa | *Phacus* | 24 |
| Euglenozoa | *Lepocinclis* | 22 |
| | Total | 75 |
| Cyanobacteria | *Phormidium* | 26 |
| | *Oscillatoria* | 19 |
| | Total | 45 |
| Chlorophyta | *Desmodesmus* | 32 |
| | Total | 32 |
| Charophyta | *Closterium* | 24 |
| | Total | 24 |
| Total | | 338 |

The ecological preferences of algae and cyanobacteria species inhabiting the water bodies of Israel are presented in Table 3 and S1. Each group of indicators was considered

separately to assess the significance of water quality bioindication in Israel. Species that predictably and similarly respond to environmental variables are used as bioindicators of these variables to reflect the response of aquatic ecosystems to eutrophication, pH (acidification), salinity, and organic pollution [14,88]. In this study, the collected indicator data were used for the ecological analysis [89,99] of water bodies in Israel as a whole (Table S1). Now we can assess the total number of indicator species distribution over ecological groups to reveal water quality in the waterbodies of Israel as a whole due to a response to climatic and anthropogenic stress influence. Figures 2–4 show that STDEV line cut off indicator groups of benthic, planktonic–benthic, and planktonic inhabitants (Figure 2a) preferred mainly temperate temperature waters but survive in a wide range of temperatures (Figure 2b), saturated by oxygen (Figure 2c), low alkaline (Figure 2d), and the broad spectrum of water pH (Figure 3a) with medium salinity (Figure 3b). Waters of Class 2 and 3 prevailed in Israel (Figure 4a,b), where algae and cyanobacteria species prefer an autotrophic type of nutrition (Figure 4c). As a whole, waters in Israel are inhabited by species that are indicators of two types of trophic state: (1) oligo- to oligo-mesotraphentic, and (2) meso-eutraphentic to eutraphentic (Figure 4d).

**Table 3.** The number of indicator species of algae and cyanobacteria in water habitats of Israel by ecological groups for the indicated environmental variables (bold). The ecological groups in each indicated parameter of the environment are arranged in order of increasing value of the indicated variable.

| Variable | No of Species | Variable | No of Species |
|---|---|---|---|
| **Habitat** | | **Saprobity indices** | |
| P | 348 | Class 1 | 49 |
| P-B | 476 | Class 2 | 351 |
| B | 603 | Class 3 | 470 |
| Ep | 59 | Class 4 | 99 |
| S | 74 | Class 5 | 11 |
| **Temperature** | | **Saprobity groups** | |
| cool | 23 | x–0.0 | 25 |
| temp | 64 | x-o–0.4 | 27 |
| eterm | 48 | o-x–0.6 | 29 |
| warm | 56 | x-b–0.8 | 35 |
| **Oxygenation** | | o–1.0 | 179 |
| $H_2S$ | 9 | o-b–1.4 | 105 |
| st | 314 | x-a–1.55 | 2 |
| st-str | 521 | b-o–1.6 | 74 |
| str | 79 | o-a–1.8 | 87 |
| aer | 51 | b–2.0 | 254 |
| ae | 12 | b-a–2.4 | 55 |
| **pH** | | a-o–2.6 | 60 |
| acf | 45 | b-p–2.8 | 3 |
| neu | 6 | a–3.0 | 37 |
| ind | 221 | a-b–3.6 | 7 |
| alf | 282 | p-a–4.0 | 1 |
| alb | 15 | i > 4.0 | 1 |
| **Salinity** | | **Trophy** | |
| hb | 34 | ot | 98 |
| i | 495 | o-m | 112 |
| hl | 106 | m | 58 |
| mh | 86 | me | 130 |
| ph | 23 | e | 109 |
| hlbnt | 2 | o-e | 25 |

**Table 3.** *Cont.*

| Variable | No of Species | Variable | No of Species |
|---|---|---|---|
| **Type of Nutrition** | | he | 8 |
| ats | 75 | **Watanabe** | |
| ate | 102 | sx | 73 |
| hne | 16 | es | 149 |
| hce | 8 | sp | 30 |

Note. Substrate preferences (P—planktonic, P-B—plankto-benthic, B—benthic, Ep—epiphyte, S—soil); temperature preferences (cool—cool water, temp—temperate, eterm—eurythermic, warm—warm water); oxygenation and streaming ($H_2S$—sulfides resistant; st—standing water, str—streaming water, st-str—low streaming water, aer—aerophiles; ae—aerophites); pH preferences groups (pH) according to Hustedt (1957) [79,93]: (alb—alkalibiontes; alf—alkaliphiles, ind—indifferent; acf—acidophiles; neu—neutrophiles as a part of pH-indifferent taxa); salinity ecological groups according to Hustedt (1938–1939) [80,94]: (hb—oligohalobes-halophobes, i—oligohalobes-indifferent, mh—mesohalobes, hl—halophiles; ph—polyhalobes; hlbnt—halobionts); self-purification zone with index of saprobity (x/0.0—xenosaprobe; x-o/0.4—xeno-oligosaprobe; o-x/0.6—oligo-xenosaprobe; o/1.0—oligosaprobe; o-b/1.4—oligo-betamesosaprobe; x-a/0.55—xeno- to alphamesosaprobe; b-o/1.6—beta-oligosaprobe; o-a/1.8—oligo-alphamesosaprobe; b/2.0—betamesosaprobe; b-a/2.4—beta-alphamesosaprobe; a-o/2.6—alpha-oligosaprobe; b-p/2.8—betapolysaprobe; a/3.0—alphamesosaprobe; a-p/3.4—alphapolysaprobe; a-b/3.6—alpha-betamesosaprobe; p-a/4.0—poly-alphamesosaprobe; i/>4.0—i-eusaprobe); organic pollution indicators according to Watanabe et al. (1986) [95]: sx—saproxenes; es—eurysaprobes; sp—saprophiles; nitrogen uptake metabolism (Aut-Het) [9]: ats—nitrogen-autotrophic taxa, tolerating very small concentrations of organically bound nitrogen; ate—nitrogen-autotrophic taxa, tolerating elevated concentrations of organically bound nitrogen; hne—facultative nitrogen-heterotrophic taxa, needing periodically elevated concentrations of organically bound nitrogen; hce—obligate nitrogen-heterotrophic taxa, needing continuously elevated concentrations of organically bound nitrogen; trophic state indicators [9]: (ot—oligotraphentic; o-m—oligomesotraphentic; m—mesotraphentic; me—mesoeutraphentic; e—eutraphentic; he—hypereutraphentic; o-e—oligo- to eutraphentic (hypereutraphentic)).

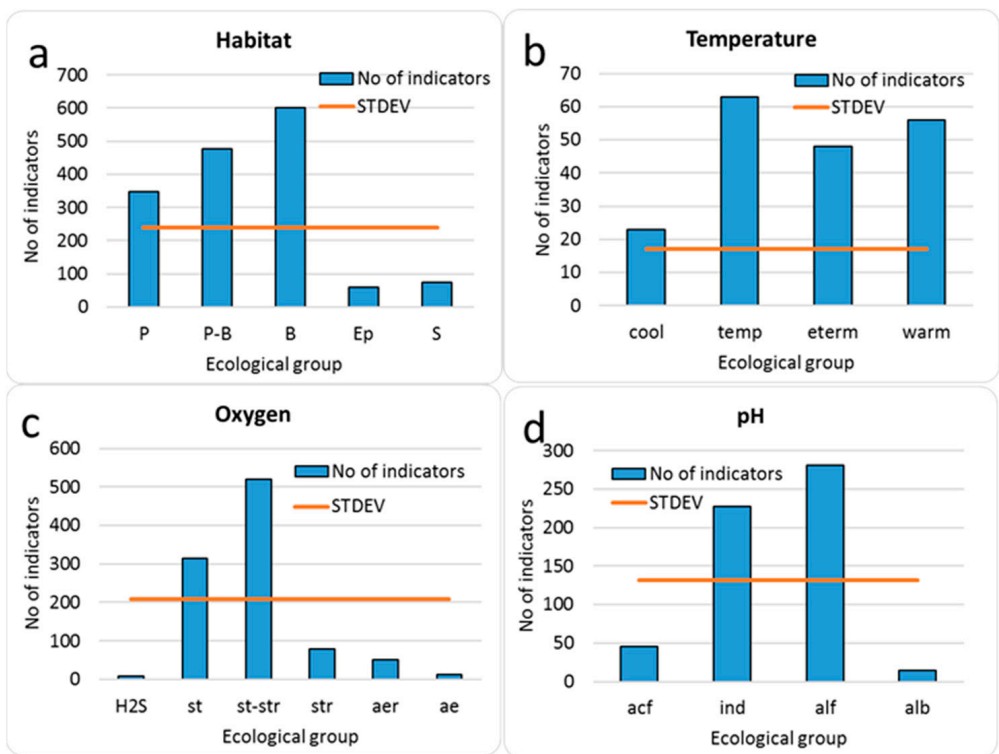

**Figure 2.** Distribution of species-indicator in groups of preferences of habitat: (P—planktonic, P-B—plankto-benthic, B—benthic, Ep—epiphyte, S—soil) (**a**). Temperature: (cool—cool water, temp—temperate, eterm—eurythermic, warm—warm water) (**b**). Oxygen: ($H_2S$—sulfides resistant; st—standing water, str—streaming water, st-str—low streaming water, aer—aerophiles; ae—aerophites) (**c**). Water pH: (alb—alkalibiontes; alf—alkaliphiles, ind—indifferent; acf—acidophiles; neu—neutrophiles as a part of pH-indifferent taxa) (**d**). STDEV: standard deviation line. Abbreviation also as in Table 3.

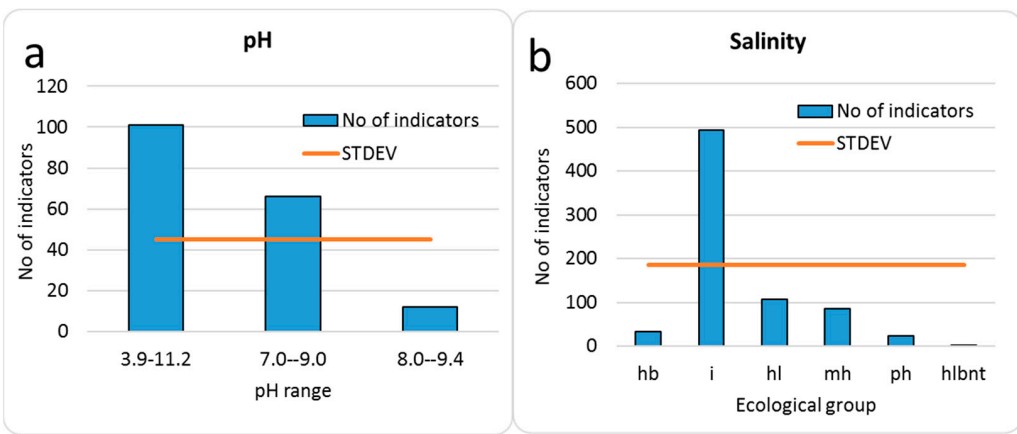

**Figure 3.** Distribution of species-indicator in groups of preferences of water pH range in which species occurred (**a**). Salinity (hb—oligohalobes-halophobes, i—oligohalobes-indifferent, mh—mesohalobes, hl—halophiles; ph—polyhalobes; hlbnt-halobionts) (**b**). STDEV: standard deviation line. Abbreviation also as in Table 3.

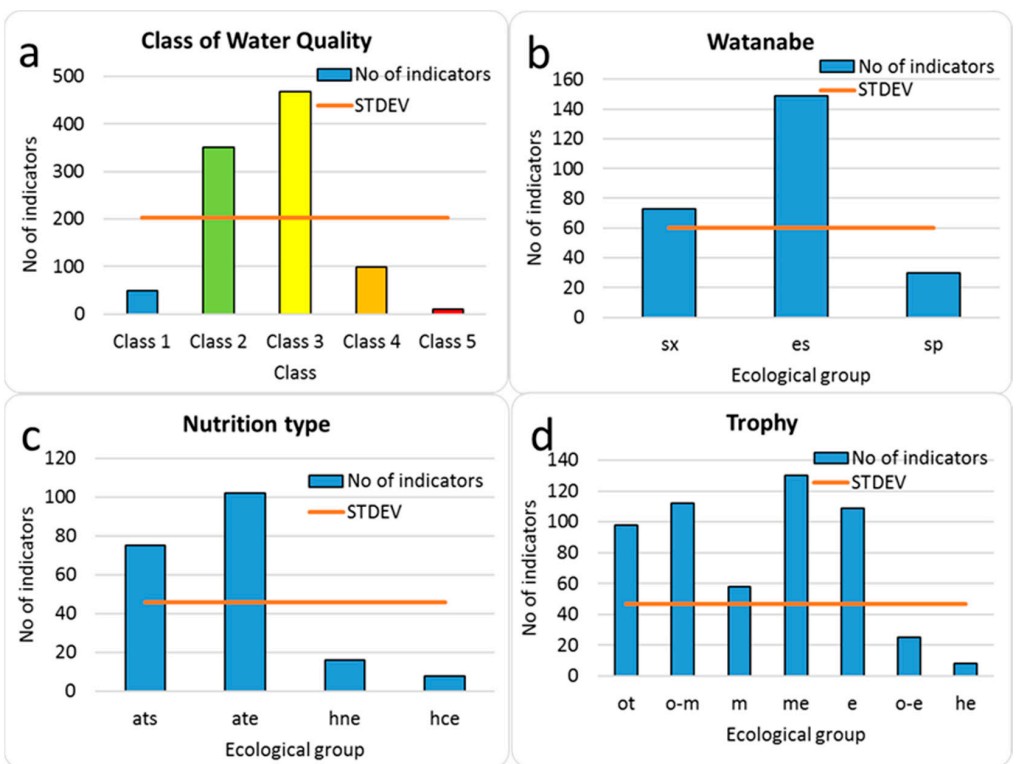

**Figure 4.** Distribution of species-indicator in groups of preferences of water quality class in the EU FWD color ranges (**a**). Organic pollution according to Watanabe (sx—saproxenes; es—eurysaprobes; sp—saprophiles) (**b**). Nutrition type as nitrogen uptake metabolism (ats—nitrogen-autotrophic taxa, tolerating very small concentrations of organically bound nitrogen; ate—nitrogen-autotrophic taxa, tolerating elevated concentrations of organically bound nitrogen; hne—facultative nitrogen-heterotrophic taxa, needing periodically elevated concentrations of organically bound nitrogen; hce—obligate nitrogen-heterotrophic taxa, needing continuously elevated concentrations of organically bound nitrogen) (**c**). Trophic state (ot—oligotraphentic; o-m—oligomesotraphentic; m—mesotraphentic; me—mesoeutraphentic; e—eutraphentic; he—hypereutraphentic; o-e—oligo-to eutraphentic (hypereutraphentic)) (**d**). STDEV: standard deviation line. Abbreviation also as in Table 3.

The experience of assessing the impact of pollution on fresh and brackish water bodies in Israel was represented in a recent study [20]. Furthermore, indicator algae have been used to assess water quality in many continental water bodies in Israel due to regional climate change [56]. Therefore, revealed algae and cyanobacteria flora with species bioindicator properties can be used in the Israeli water quality monitoring as implemented in the EU [11,12], but now in Israel, the chemical variables are monitored only. In any case, the experience of using bioindicators to assess water quality in continental water bodies of Israel is rather large [20]. It can help assess the ecosystems state and reveal pollution sources in each aquatic object [56] and future water quality monitoring.

In closely related regions such as Turkey [4], with 2030 taxa, and Iraq, with 2647 species [5], and Georgia [6], the algal floras are, as a whole, enriched by non-diatom algae. The well-studied flora of Ukraine contains 6583 algae and cyanobacteria taxa with prevailing diatoms, and half of the list were indicators of the environmental variables [7], which can be used in the study of evolutionary dynamics and as indicators of climate change [56,100]. Algae and cyanobacteria floras of close related regions to Israel, such as Lebanon, Egypt, and Jordan, do not yet have the checklists and they stay in the initial stage of biodiversity research. We collect the biodiversity lists from Western Eurasia, and comparative floristic analysis will be the next research stage.

## 4. Conclusions

The list of algae and cyanobacteria in the continental waters of Israel not only summarized its diversity with advances in XXI century research, but also first represented each species' ecological preferences. We conclude that species diversity during the past 20 years has raised from 1261 to 1628 species, which belong to 14 taxonomic phyla. All identified species can be used as bioindicators of water quality. The ecological properties of indicators are associated with ten environmental variables. In addition, the change in the ecological state affected many of the water bodies mentioned in the published articles, and some of the water bodies were either rebuilt or destroyed. The impact of pollution on freshwater and brackish water ecosystems of Israel can be assessed with the help of modern bioindication methods in the water quality monitoring system that now represents the chemical data only.

**Supplementary Materials:** The following supporting information can be downloaded at: https://www.mdpi.com/article/10.3390/d14050328/s1, Table S1. List of algae and cyanobacteria in continental waterbodies of Israel (1898–2022) with species ecological preferences.

**Author Contributions:** Conceptualization, S.B.; methodology, S.B.; software, S.B.; validation, S.B. and T.S.; formal analysis, S.B.; investigation, S.B. and T.S.; resources, S.B.; data curation, S.B. and T.S.; writing—original draft preparation, S.B. and T.S.; writing—review and editing, S.B. and T.S.; visualization, S.B.; supervision, S.B.; project administration, S.B. All authors have read and agreed to the published version of the manuscript.

**Funding:** This research received no external funding.

**Institutional Review Board Statement:** Not applicable.

**Data Availability Statement:** Not applicable.

**Acknowledgments:** This work was partly supported by the Israeli Ministry of Aliyah and Integration. We are also thankful to Olena Cherniavska for technical assistance.

**Conflicts of Interest:** The authors declare no conflict of interest.

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
