# Peer review of "Flora of Algae and Cyanobacteria of Continental Waters of Israel in the XXI Century: Taxonomy, Autecology and Water Quality Indicators"

_diversity, doi:10.3390/d14050328_

Round 1
Reviewer 1 Report
Dear Authors,
Is the manuscript clear, relevant for the field and presented in a well-structured manner?
I think that the study is valuable and relevant to the field. In general, the article is mostly clearly presented and the structure makes sense to me.
Having written this, I'd also like to mention the some difficulties I encountered.
Specific comments:
Line 29-30, 30-32, 46-48, 52-53: citations are needed
In Introduction chapter, there is no references to similar studies conducted in other countries. This could be more precise. In which years the research was conducted? See Line 226 and Line 227.
Line 73: “…modern system…” I am confused (see Line 90); which modern system? which keys were used by the authors to identify of species?
Line 115: The classification follows the 2011 edition of “The Freshwater Algal Flora of the British Isles” and recognises 15 phyla (see Table 1). It differs in part from AlgaeBase, which to a large extent follows the consensus classification presented in Ruggiero et al. 2015. No explanation (citation) is available. The classification (using in Table 1) is not clarified.
Line 168: why authors using “preferences groups according Hustedt (1957)’?

Author Response
Dear Editor and the Reviewer 1,
Thank you for comments and recommendations. I following it and corrected my ms. Please find my point by point responses below.
With best regards,
Sophia Barinova
Reviewer 1 respobses.
Dear Authors,
Is the manuscript clear, relevant for the field and presented in a well-structured manner?
I think that the study is valuable and relevant to the field. In general, the article is mostly clearly presented and the structure makes sense to me.
Having written this, I'd also like to mention the some difficulties I encountered.
Specific comments:
Line 29-30, 30-32, 46-48, 52-53: citations are needed
Response: text reweighted, citation added.
The adaptation level of species determines the species composition in a given aquatic
Response: citation added
In Introduction chapter, there is no references to similar studies conducted in other countries. This
could be more precise. In which years the research was conducted? See Line 226 and Line 227.
Response: Similar but not the same work has been done for Ukraine, ref. 78. As stated in the MM, the work of specific authors on Israeli algae refers to two periods 1898-2000 and 2001-2022. As regards the floras of Turkey, Iraq, Georgia, and Ukraine, data were collected over a number of years and then combined into their published algae floras, refs 90, 91, 92, and 78. Collection of published material from the region is a specific work of compiling flora.
Line 73: “…modern system…” I am confused (see Line 90); which modern system? which keys were used by the authors to identify of species?
Response: As stated above in Materials and Methods, the classification follows the modern system using algaebase.org [86]. The handbooks used by the authors of specific papers are indicated in the cited specific papers.
Line 115: The classification follows the 2011 edition of “The Freshwater Algal Flora of the British Isles” and recognises 15 phyla (see Table 1). It differs in part from AlgaeBase, which to a large extent follows the consensus classification presented in Ruggiero et al. 2015. No explanation (citation) is available. The classification (using in Table 1) is not clarified.
Response: As stated above in Materials and Methods, Classification follows the modern system using algaebase.org [86]
Line 168: why authors using “preferences groups according Hustedt (1957)’?
Response: This is a widely used system, and most importantly, it covers the entire range of the indicated indicator existing in nature, with a fairly fractional division into indicator groups.

Reviewer 2 Report
You did a theoretical work based on collected data through the years. You tried to categorize your data according to selected parameters. It is a good work. But at the end the reader has not a take-away conclusion about the present biological - ecological state of Israel's water bodies base on your floristic data. Do you think it as a good idea to add 1 or 2 phrases at your conclusion about the present as compared to the past on this issue?
Some corrections. Lines 91,92. Please write properly something is wrong in syntax.
Line 100. Amend syntax error.
Lines 109, 110, take care of the parentheses.
Line 118. Delete 3.1.
In Figure 1b the labels on the X-axis are oversized, cannot be discerned, reduce them.
Author Response
Dear Editor and the Reviewer 2,
Thank you for comments and recommendations. I following it and corrected my ms. Please find my point by point responses below.
With best regards,
Sophia Barinova
Reviewer 2 responses.
You did a theoretical work based on collected data through the years. You tried to categorize your data according to selected parameters. It is a good work. But at the end the reader has not a take-away conclusion about the present biological - ecological state of Israel's water bodies base on your floristic data. Do you think it as a good idea to add 1 or 2 phrases at your conclusion about the present as compared to the past on this issue?
Response: added
Some corrections. Lines 91,92. Please write properly something is wrong in syntax.
Response: corrected
Line 100. Amend syntax error.
Response: corrected
Lines 109, 110, take care of the parentheses.
Response: corrected
Line 118. Delete 3.1.
Response: done
In Figure 1b the labels on the X-axis are oversized, cannot be discerned, reduce them.
Response: done, figure replaced
